# Deep evolutionary analysis reveals the design principles of fold A glycosyltransferases

Rahil Taujale[1,2], Aarya Venkat[3], Liang-Chin Huang[1], Zhongliang Zhou[4], Wayland Yeung[1], Khaled M Rasheed[4], Sheng Li[4], Arthur S Edison[1,2,3], Kelley W Moremen[2,3], Natarajan Kannan[1,3]*

[1]Institute of Bioinformatics, University of Georgia, Athens, Georgia; [2]Complex Carbohydrate Research Center, University of Georgia, Athens, Georgia; [3]Department of Biochemistry and Molecular Biology, University of Georgia, Athens, Georgia; [4]Department of Computer Science, University of Georgia, Athens, Georgia

**Abstract** Glycosyltransferases (GTs) are prevalent across the tree of life and regulate nearly all aspects of cellular functions. The evolutionary basis for their complex and diverse modes of catalytic functions remain enigmatic. Here, based on deep mining of over half million GT-A fold sequences, we define a minimal core component shared among functionally diverse enzymes. We find that variations in the common core and emergence of hypervariable loops extending from the core contributed to GT-A diversity. We provide a phylogenetic framework relating diverse GT-A fold families for the first time and show that inverting and retaining mechanisms emerged multiple times independently during evolution. Using evolutionary information encoded in primary sequences, we trained a machine learning classifier to predict donor specificity with nearly 90% accuracy and deployed it for the annotation of understudied GTs. Our studies provide an evolutionary framework for investigating complex relationships connecting GT-A fold sequence, structure, function and regulation.

*For correspondence:
nkannan@uga.edu

## Introduction

Complex carbohydrates make up a large bulk of the biomass of any living cell and play essential roles in biological processes ranging from cellular interactions, pathogenesis, immunity, quality control of protein folding and structural stability (*Varki and Gagneux, 2019*). Biosynthesis of complex carbohydrates in most organisms is carried out by a large and diverse family of Glycosyltransferases (GTs) that transfer sugars from activated donors such as nucleotide diphosphate and monophosphate sugars or lipid linked sugars to a wide range of acceptors that include saccharides, lipids, nucleic acids and metabolites. Nearly 1% of protein coding genes in the human genome, and more than 2% of the *Arabidopsis* genome, are estimated to be GTs. GTs have undergone extensive variation in primary sequence and three-dimensional structure to catalyze the formation of glycosidic bonds between diverse donor and acceptor substrates. However, an incomplete understanding of the relationships connecting sequence, structure, function and regulation presents a major bottleneck in understanding pathogenicity, metabolic and neurodegenerative diseases associated with abnormal GT functions (*Ryan et al., 2019*; *Day et al., 2012*).

Structurally, GTs adopt one of three folds (GT-A, -B or -C) with the GT-A Rossmann like fold being the most common (*Figure 1*, *Figure 1—source data 1*). The GT-A fold is characterized by alternating β-sheets and α-helices (α/β/α sandwich) found in most nucleotide binding proteins (*Breton et al., 2012*). The majority of GT-A fold enzymes are metal dependent and conserve a DxD motif in the active site that helps coordinate the metal ion and the nucleotide sugar. Currently, 110

**eLife digest** Carbohydrates are one of the major groups of large biological molecules that regulate nearly all aspects of life. Yet, unlike DNA or proteins, carbohydrates are made without a template to follow. Instead, these molecules are built from a set of sugar-based building blocks by the intricate activities of a large and diverse family of enzymes known as glycosyltransferases.

An incomplete understanding of how glycosyltransferases recognize and build diverse carbohydrates presents a major bottleneck in developing therapeutic strategies for diseases associated with abnormalities in these enzymes. It also limits efforts to engineer these enzymes for biotechnology applications and biofuel production.

Taujale et al. have now used evolutionary approaches to map the evolution of a major subset of glycosyltransferases from species across the tree of life to understand how these enzymes evolved such precise mechanisms to build diverse carbohydrates. First, a minimal structural unit was defined based on being shared among a group of over half a million unique glycosyltransferase enzymes with different activities. Further analysis then showed that the diverse activities of these enzymes evolved through the accumulation of mutations within this structural unit, as well as in much more variable regions in the enzyme that extend from the minimal unit.

Taujale et al. then built an extended family tree for this collection of glycosyltransferases and details of the evolutionary relationships between the enzymes helped them to create a machine learning framework that could predict which sugar-containing molecules were the raw materials for a given glycosyltransferase. This framework could make predictions with nearly 90% accuracy based only on information that can be deciphered from the gene for that enzyme.

These findings will provide scientists with new hypotheses for investigating the complex relationships connecting the genetic information about glycosyltransferases with their structures and activities. Further refinement of the machine learning framework may eventually enable the design of enzymes with properties that are desirable for applications in biotechnology.

GT families have been catalogued in the Carbohydrates Active Enzymes (CAZy) database (accessed in February 2020) (*Lombard et al., 2014*). These families can be broadly classified into two categories based on their mechanism of action and the anomeric configuration of the glycosidic product relative to the sugar donor, namely, inverting or retaining (*Figure 1*). Inverting GTs generally employ an $S_N2$ single displacement reaction mechanism that results in inversion of anomeric configuration for the product. In contrast, retaining GTs are believed to employ a dissociative $S_Ni$-type mechanism, where the anomeric configuration of the product is retained (*Moremen and Haltiwanger, 2019*; *Lairson et al., 2008*). While the sequence basis for inverting and retaining mechanisms is not well understood, most inverting GT-As have a conserved Asp or Glu within a xED motif that serves as the catalytic base to deprotonate the incoming nucleophile of the acceptor, and initiate nucleophilic attack with direct displacement of the phosphate leaving group (*Lairson et al., 2008*; *Gloster, 2014*). Retaining GT-As bind the sugar donor similarly to the inverting enzymes, but shift the position of the acceptor nucleophile to attack the anomeric carbon from an obtuse angle using a phosphate oxygen of the sugar donor as the catalytic base and employ a dissociative mechanism that retains the anomeric linkage for the resulting glycosidic product (*Moremen and Haltiwanger, 2019*). Such mechanistic diversity of GTs is further illustrated by recent crystal structures of GTs bound to acceptor and donor substrates which show that different acceptors are accommodated in the active site through variable loop regions emanating from the catalytic core (*Moremen and Haltiwanger, 2019*; *Ramakrishnan and Qasba, 2010a*; *Kadirvelraj et al., 2018*; *Gordon et al., 2006*). However, whether these observations hold for the entire super-family is not known because of the lack of structural information for the vast number of GTs.

The wealth of sequence data available on GTs provides an opportunity to infer underlying mechanisms through deep mining of large sequence datasets. In this regard, the CAZY database serves as a valuable resource (*Lombard et al., 2014*) for generating new functional hypotheses by classifying GT enzymes into individual families based on overall sequence similarity. However, a broader understanding of how these enzymes evolved to recognize diverse donor and acceptor substrates requires a global comparison of diverse GT-A fold enzymes. Such comparisons are currently a challenge due

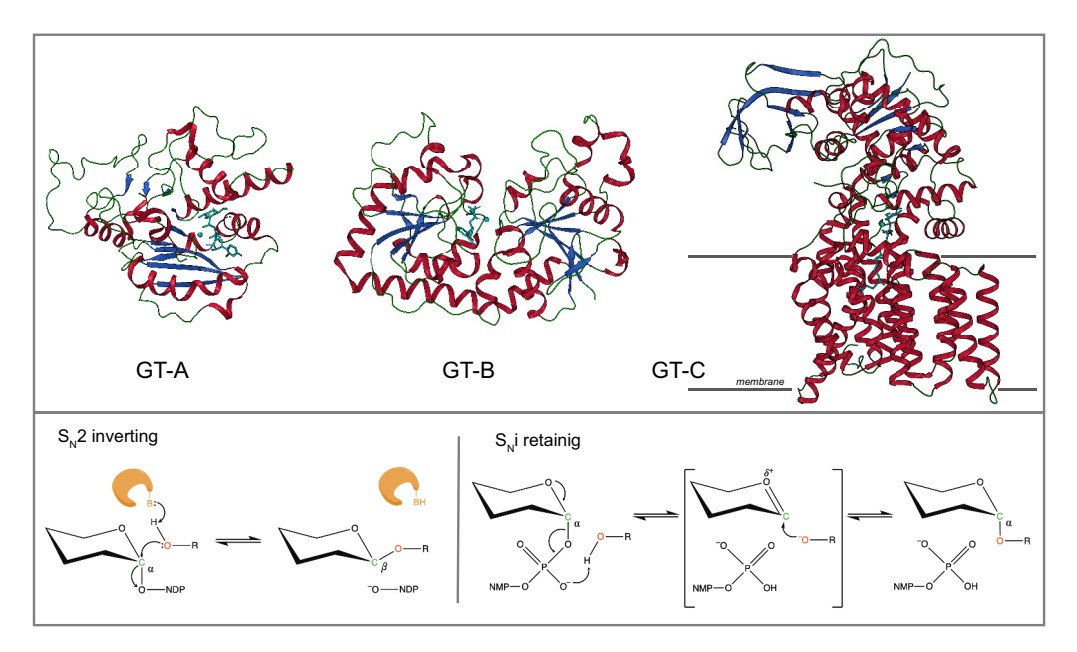

**Figure 1.** Glycosyltransferase (GT) folds and mechanisms. Top: The three representative structural folds of GTs. The GT-A fold is characterized by a single globular domain that contains a α/β/α Rossmann nucleotide binding domain (shown 2rj7;GT6). The GT-B fold enzymes are usually metal independent and contain two α/β/α domains separated by a flexible linker region with the substrate binding cleft in between (shown 1jg7;GT63). The GT-C fold enzymes are hydrophobic integral membrane proteins, generally use lipid phosphate linked sugar donors and have multiple transmembrane helices (shown 6gxc; GT66). Bottom: The mechanism of sugar transfer employed by GTs. Inverting GTs follow a direct displacement $S_N$-2-like mechanism that results in an inverted anomeric configuration. The mechanism for retaining GTs is still under debate although recently a same side $S_Ni$-type reaction has been proposed where the donor phosphate oxygen acts as a catalytic base and deprotonates the acceptor hydroxyl facilitating a same side attack, that results in the retention of anomeric configuration. The enzyme and catalytic base B are shown in orange. A generic hexose with α-linkage to a nucleoside diphosphate is used. Other mechanisms possibly employed by GTs is discussed in detail in M.

The online version of this article includes the following source data for figure 1:

**Source data 1.** List of CAZy GT families.

to limited sequence similarity between families and the lack of a phylogenetic framework to detect evolutionary events associated with GT functional specialization. Previous efforts to investigate GT evolution have largely focused on individual families or pathways (*Taujale and Yin, 2015*; *Lombard, 2016*) and have not explicitly addressed the challenge of mapping the evolution of functional diversity across families.

Here through deep mining of over half a million GT-A fold-related sequences from diverse organisms, and application of specialized computational tools developed for the study of large gene families (*Kannan et al., 2007*; *Kwon et al., 2019*), we define a common core shared among diverse GT-A fold enzymes. Using the common core features, we generate a phylogenetic framework for relating functionally diverse enzymes and show that inverting and retaining mechanisms emerged independently multiple times during evolution. We identify convergent modes of substrate recognition in evolutionarily divergent families and pinpoint sequence and structural features associated with functional specialization. Finally, based on the evolutionary and structural features gleaned from a broad analysis of diverse GT-A fold enzymes, we develop a machine learning (ML) framework for predicting donor specificity with nearly 90% accuracy. We predict donor specificity for uncharacterized GT-A enzymes in diverse model organisms and provide testable hypotheses for investigating the relationships connecting GT-A fold structure, function and evolution.

## Results

### An ancient common core shared among diverse GT-A fold enzymes

To define common features shared among diverse GT-A fold enzymes, we generated a multiple sequence alignment of over 600,000 GT-A fold related sequences in the non-redundant (NR) sequence database (*Pruitt et al., 2007*) using curated multiple-aligned profiles of diverse GTs. The alignment profiles were curated using available crystal structures (Materials and methods) (*Neuwald, 2009*). The resulting alignment revealed a GT-A common core consisting of 231 aligned positions. These aligned positions are referred to throughout this analysis and are mapped to representative structures in *Supplementary file 2*. The common core is defined by eight β sheets and six α helices, including three β sheets and α helices from the N-terminal Rossmann fold (*Figure 2A,B*).

Quantification of the evolutionary constraints imposed on the common core reveal twenty residues shared among diverse GT-A fold families. These include the DxD and the xED motif residues

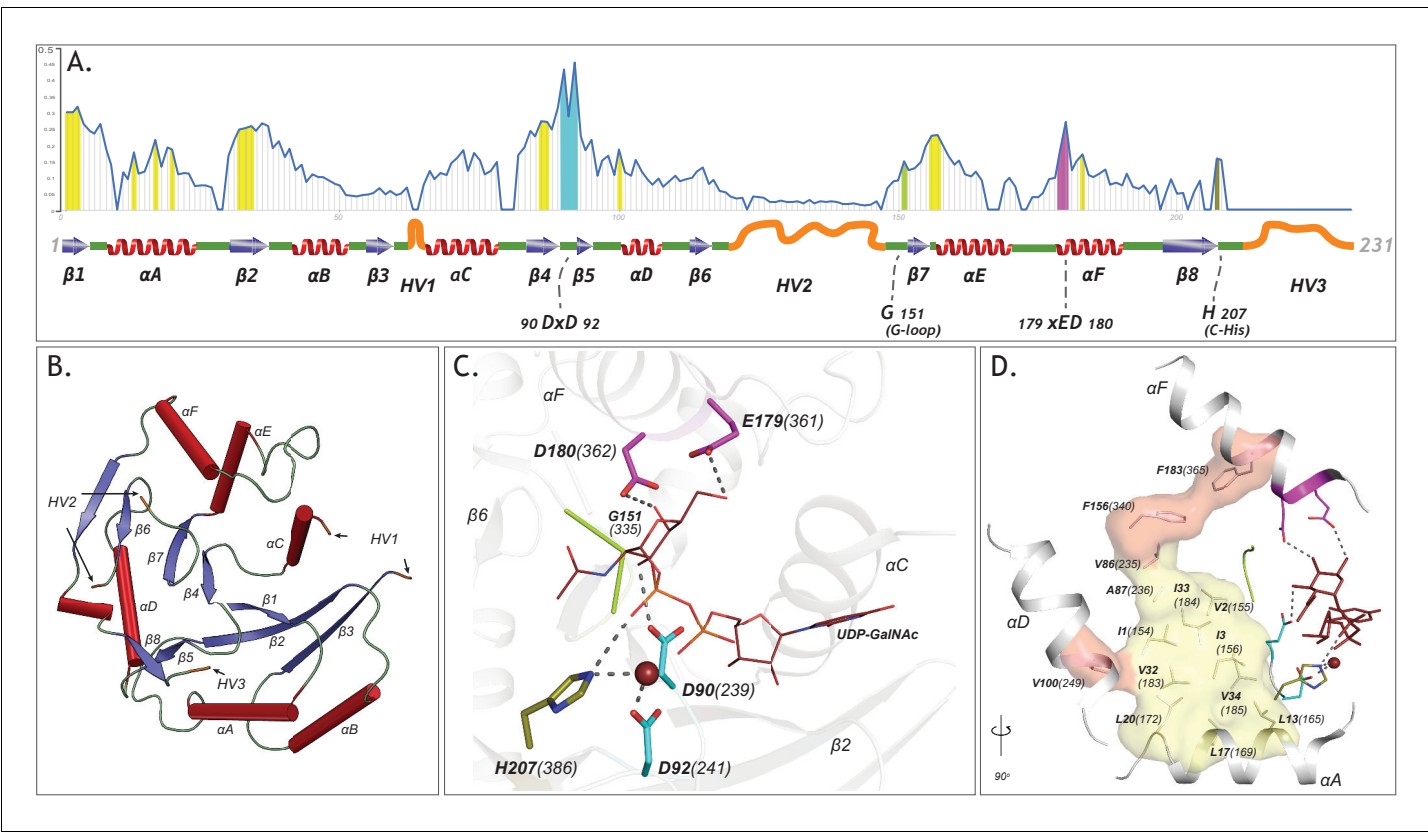

**Figure 2.** The GT-A common core and its elements. (**A**) Plot showing the schematics of the GT-A common core with 231 aligned positions. Conserved secondary structures (red α-helices, blue β-sheets, green loops) and hypervariable regions (HVs)(orange) are shown. Conservation score for each aligned position is plotted in the line graph above the schematics. Evolutionarily constrained regions in the core: the hydrophobic positions (yellow) and the active site residues (DxD: Cyan, xED: Magenta, G-loop: green, C-His: olive) are highlighted above the positions. (**B**) The conserved secondary structures and the location of HVs are shown in the N-terminal GT2 domain of the multidomain chondroitin polymerase structure from *E. coli*(PDB: 2z87) that is used as a prototype as it displays closest similarity to the common core consensus. (**C**) Active site residues of the prototypic GT-A structure. Metal ion and donor substrate are shown as a brown sphere and sticks, respectively. (**D**) Architecture of the hydrophobic core (Yellow: core conserved in all Rossmann fold containing enzymes, Red: core elements present only in the GT-A fold). Residues are labeled based on their aligned positions. Numbers within parentheses indicate their position in the prototypic (PDB: 2z87) structure.

The online version of this article includes the following figure supplement(s) for figure 2:

**Figure supplement 1.** Structure based sequence alignment showing the hydrophobic residue positions present across a collection of Rossmann fold like enzymes.

**Figure supplement 2.** Changes in the extended hydrophobic core residues in selected retaining families.

**Figure supplement 3.** Comparison of structures for HV regions across GT-A families.

involved in catalytic functions, and other residues not typically associated with catalysis (*Figure 2A*) such as the conserved glycine at aligned position 151 (G335 in 2z87) in the flexible G-loop and a histidine residue (H386 in 2z87) in the C-terminal tail at aligned position 207, henceforth referred to as the C-His. Residues from the G-loop in some families, such as the blood ABOs (GT6) and glucosyl-3-phosphoglycerate synthases (GpgS; GT81), contribute to donor binding (*Patenaude et al., 2002*; *Empadinhas et al., 2011*). The C-His, likewise, coordinates with the metal ion and contributes to catalysis in a subset of GTs, such as polypeptide N-acetylgalactosaminyl transferases (ppGalNAcTs; GT27) and lipopolysaccharyl-$\alpha$−1,4-galactosyltransferase C (LgtC; GT8) (*Fritz et al., 2004*; *Persson et al., 2001*). The conservation of these residues across diverse GT-A fold enzymes suggest that they likely perform similar functional roles in other families as well.

The remaining core conserved residues include fourteen hydrophobic residues that are dispersed in sequence, but spatially cluster to connect the catalytic site and the Rossmann fold. Eleven out of the fourteen residues (highlighted in yellow in *Figure 2D*) are shared by other Rossmann fold proteins (*Figure 2—figure supplement 1*) suggesting a role for these residues in maintaining the overall fold. Three hydrophobic residues (V249, F340, F365; shown in red surface in *Figure 2D*), however, are unique to GT-A fold enzymes, and structurally bridge the αF helix (containing the xED motif), the αD helix and the Rossmann fold domain. Although the functional significance of this hydrophobic coupling is not evident from crystal structures, in some families (GT15 and GT55) the hydrophobic coupling between αF and the Rossmann fold domain is replaced by charged interactions (*Figure 2—figure supplement 2*). The structural and functional significance of these family specific variations are discussed below.

Our broad evolutionary analysis also reveals three hypervariable regions (HVs) extending from the common core. These include an extended loop segment connecting β3 strand and αC helix (HV1), a segment longer than 28 amino acids connecting β6 and β7 strand (HV2) and a C-terminal tail extending from the β8 strand (HV3) in the common core. These HVs, while conserved within families, display significant conformational and sequence variability across families (*Figure 2A*, *Figure 2—figure supplement 3*) and encode family-specific motifs that contribute to acceptor specificity in individual families, as discussed below.

## A phylogenetic framework relating diverse GT-A fold families

Having delineated the common core, we next sought to generate a phylogenetic tree relating diverse GT-A fold families using the core alignment. Because of the inherent challenges in the generation and visualization of large trees (*Sanderson and Driskell, 2003*), we used a representative set of GT-A fold sequences for phylogenetic analysis by first clustering the ~600,000 sequences into functional categories using a Bayesian Partitioning with Pattern Selection (BPPS) method (*Neuwald, 2014*). The BPPS method partitions sequences in a multiple sequence alignment into hierarchical sub-groups based on correlated residue patterns characteristic of each sub-group (Materials and methods). This revealed 99 sub-groups with distinctive patterns. Representative sequences across diverse phyla from these sub-groups (993 sequences, *Figure 3—source data 2*) were then used to generate a phylogenetic tree (*Figure 3*). Based on the phylogenetic placement of these sequences, we broadly define fifty-three major sub-groups, thirty-one of which correspond to CAZy-defined families (*Figure 3—source data 1*). The remaining sub-groups correspond to sub-families within larger CAZy families. In particular, we sub-classified the largest GT family in the CAZy database, GT2, into ten phylogenetically distinct sub-families. Likewise, GT8 and GT31 were classified into seven and five sub-families, respectively. These sub-families are not explicitly captured in CAZy and are annotated based on overall sequence similarity to functionally characterized members. For example, 'GT2-LpsRelated' corresponds to a sub-family within GT2 most closely related to the bacterial β−1–4-glucosyltransferases (lgtF) involved in Lipopolysaccharide biosynthesis (*Figure 3*, *Figure 3—figure supplement 1*). Such a hierarchical classification captures the evolutionary relationships between GT-A fold families/sub-families while keeping the nomenclature consistent with CAZy.

GT-A fold families and sub-families can be further grouped into clades based on shared sequence features and placement in the phylogenetic tree (*Figure 3*). For example, clade one groups four GT2 sub-families (GT2-CeS, GT2-CWR, GT2-Chitin-HAS and GT2-Bre3) with GT84 and GT21 with high confidence, as determined by bootstrap values (see *Figure 3* legends). Members of these six families are all involved in either polysaccharide or glycosphingolipid biosynthesis. Additionally, the pattern-based classification identified a conserved [QR]XXRW motif in the C-terminal HV3

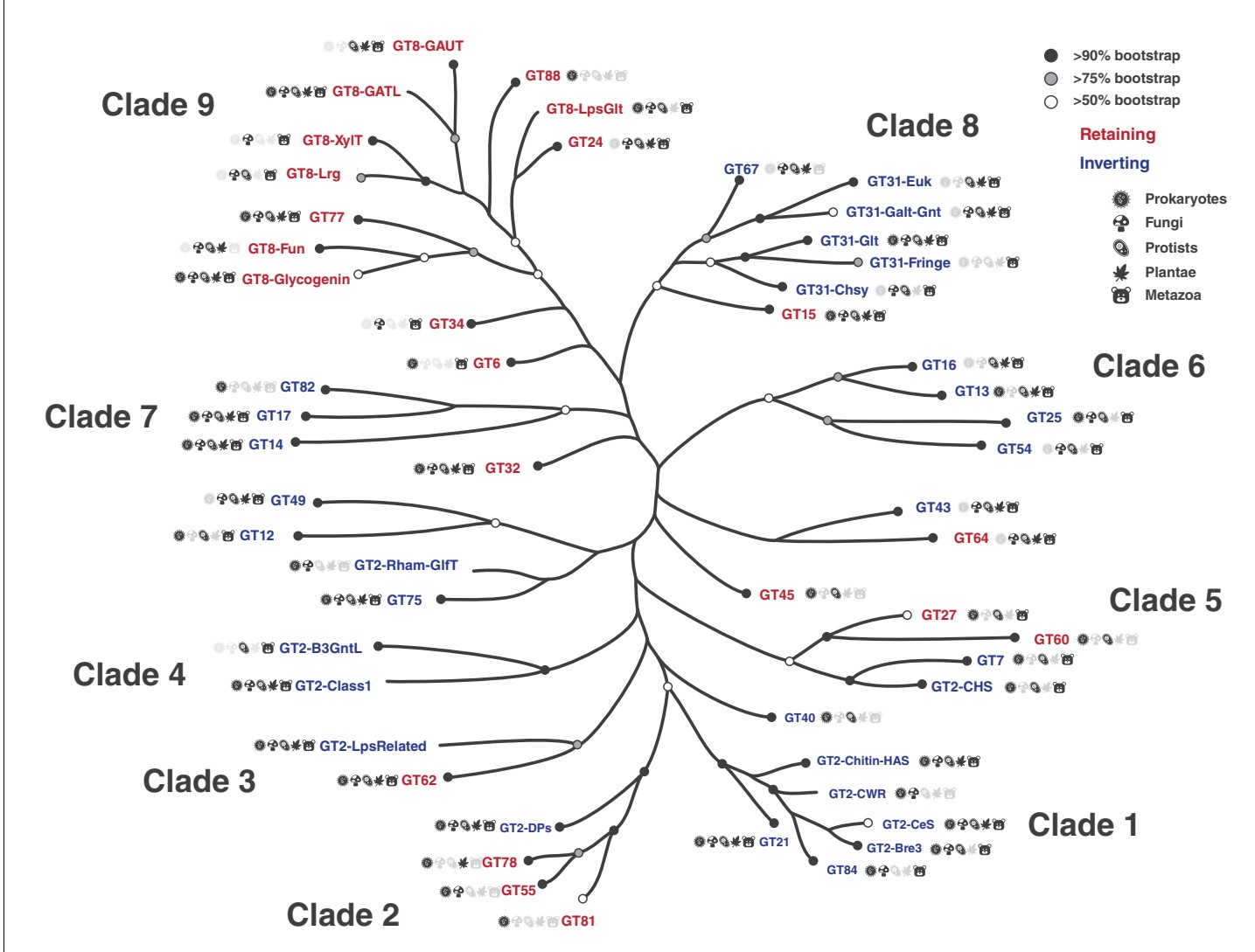

**Figure 3.** Phylogenetic tree highlighting the 53 major GT-A fold subfamilies. Tips in this tree represent GT-A sub-families condensed from the original tree for illustration. Support values are indicated using different circles. Circles at the tips indicate bootstrap support for the GT-A family clade represented by that tip. Tips missing the circles represent GT-A families that do not form a single monophyletic clade. Nodes missing circles have a bootstrap support less than 50% and are unresolved. Icon labels indicate the taxonomic diversity of that sub clade. Colors indicate the mechanism for the families (blue: Inverting, red: Retaining). This condensed tree was generated by collapsing clades to the deepest node that includes sequences from the same family. For GT-A families that did not form a monophyletic clade, the clade that included the most sequences from that family was chosen. Branch lengths may approximate the original distances, but are not drawn to scale. Detailed tree with support values, expanded nodes and scaled branch lengths are provided in *Figure 3—figure supplement 1* and in Newick format in *Figure 3—source data 4*. The family names are described in *Figure 3—source data 1*.

The online version of this article includes the following source data and figure supplement(s) for figure 3:

**Source data 1.** List of GT-A fold families and subfamilies.
**Source data 2.** The 993 representative GT-A domain sequences included in the phylogenetic analysis.
**Source data 3.** The trimmed FASTA alignment of the 231 positions of the GT-A core used for phylogeny.
**Source data 4.** The phylogenetic tree file for the 993 GT-A fold sequences in Newick format.
**Figure supplement 1.** Complete phylogenetic tree of 993 representative GT-A sequences.
**Figure supplement 2.** Clade specific conserved features in the HVs.
**Figure supplement 3.** Sankey diagram comparing topologies of phylogenetic tree with pdb and hmm based clustering of GT-A families.

(*Figure 3—figure supplement 2*) which is unique to members of this clade. The [QR]XXRW motif residues coordinate with the donor and acceptor in a bacterial cellulose synthase (from GT2-CeS family) (*Morgan et al., 2013*) and mutation of these residues in bacterial cyclic β−1,2-glucan synthetase (Cgs, GT84) abrogates activity (*Ciocchini et al., 2006*), suggesting a critical role of this motif in functional specialization of clade 1 GT-As.

The GT8 sub-families form sub-clades within the larger clade 9. For example, GT8 sequences involved in the biosynthesis of pectin components group together in the GT8-GAUT and GT8-GATL families (*Figure 3*). The human LARGE1 and LARGE2 GTs are multi-domain enzymes with two tandem GT-A domains. Their N-terminal GT-A domains fall into the GT8-Lrg subfamily that groups closely with GT8-xylosyltransferase (GT8-XylT) subfamily enzymes and places all the GT8 xylosyltransferases into a single well supported sub clade. The lipopolysaccharide α-glucosyltransferases (GT8-LpsGlt) group with the glucosyltransferases of the GT24 family, suggesting a common ancestor associated with glucose donor specificity. On the other hand, the GT8-Glycogenin sub family, which also includes members that transfer a glucose, is placed in a separate sub-clade, possibly indicating an early divergence for its unique ability to add glucose units to itself (*Alonso et al., 1995*). Clade nine members also share common sequence features associated with substrate binding that includes a lysine residue within the commonly shared KPW motif in HV3 that coordinates with the phosphate group of the donor (e.g. bacterial LgtC GT8-LpsGlt and other structures of clade nine members) (*Figure 3—figure supplement 2*).

We noticed that three out of four MGAT GT-A families responsible for the branching of N-glycans (GT13 MGAT1, GT16 MGAT2 and GT54 MGAT4) fall in the same clade (clade 6), as expected (*Figure 3*). In contrast, the fourth family, GT17 MGAT3, which adds a bisecting GlcNAc to a core β−mannose with a β−1,4 linkage, is placed in a separate clade with GT14 and GT82 (clade 7), while a fifth MGAT member creating β−1,6-GlcNAc linkages (GT18 MGAT5) is a GT-B fold enzyme (*Nagae et al., 2018*).

We further note that fifteen out of fifty-three GT-A families are found in both prokaryotes and eukaryotes. These fifteen families fall on different clades throughout the tree. GT-A families present only in prokaryotes, like GT81, GT82 and GT88, are also spread out in different clades (*Figure 3*). Similarly, other GT-A families that are present within restricted subsets of taxonomic groups (like GT40 and GT60 present only in prokaryotes and protists) are also scattered throughout the tree. These observations suggest that the divergence of most GT-A families predates the separation of prokaryotes and eukaryotes.

## Multiple evolutionary lineages for inverting and retaining mechanisms

To obtain insights into the evolution of catalytic mechanism, we annotated the phylogenetic tree based on known mechanisms of action (inverting or retaining). Inverting GTs are colored in blue in the phylogenetic tree, while retaining GTs are colored in red (*Figure 3*). The dispersion of inverting and retaining families in multiple clades suggests that these catalytic mechanisms emerged independently multiple times during GT-A fold evolution. We find that natural perturbations in the catalytic base residue, an important distinction between the inverting and retaining mechanisms, correlates well with these multiple emergences across the tree. The residue that acts as a catalytic base for inverting GTs (aspartate within the xED motif, xED-Asp) is variable across the retaining families consistent with its lack of role in the retaining $S_N$i mechanism (*Moremen and Haltiwanger, 2019*). In the inverting families, the xED-Asp is nearly always conserved and appropriately positioned to function as a catalytic base (*Figure 4A*), though some exceptions have been noted (*Moremen and Haltiwanger, 2019*; *Gandini et al., 2017*). Out of the five clades grouping inverting and retaining families, inverting families in three of these clades do not conserve the xED-Asp (GT2-DPs, GT2-LpsRelated and GT43). The heterogeneous nature of this residue in these families suggests that change of the catalytic base residue could be a key event in the transition between inverting and retaining mechanisms. Unlike families that conserve the xED-Asp, these families achieve inversion of stereochemistry through alternative modes that may relieve the constraints necessary to conserve the xED-Asp. For example, in GT43, the Asp base is replaced by a glutamate residue, which shifts the reaction center by one carbon bond (*Moremen and Haltiwanger, 2019*). Further, the dolichol phosphate transferases (DPMs and DPGs) in the GT2-DP family, which lack the xED-Asp entirely, transfer sugars to a negatively charged acceptor substrate (a phosphate group) and thus do not need a catalytic base to initiate nucleophilic attack (*Gandini et al., 2017*). Other GT-A inverting

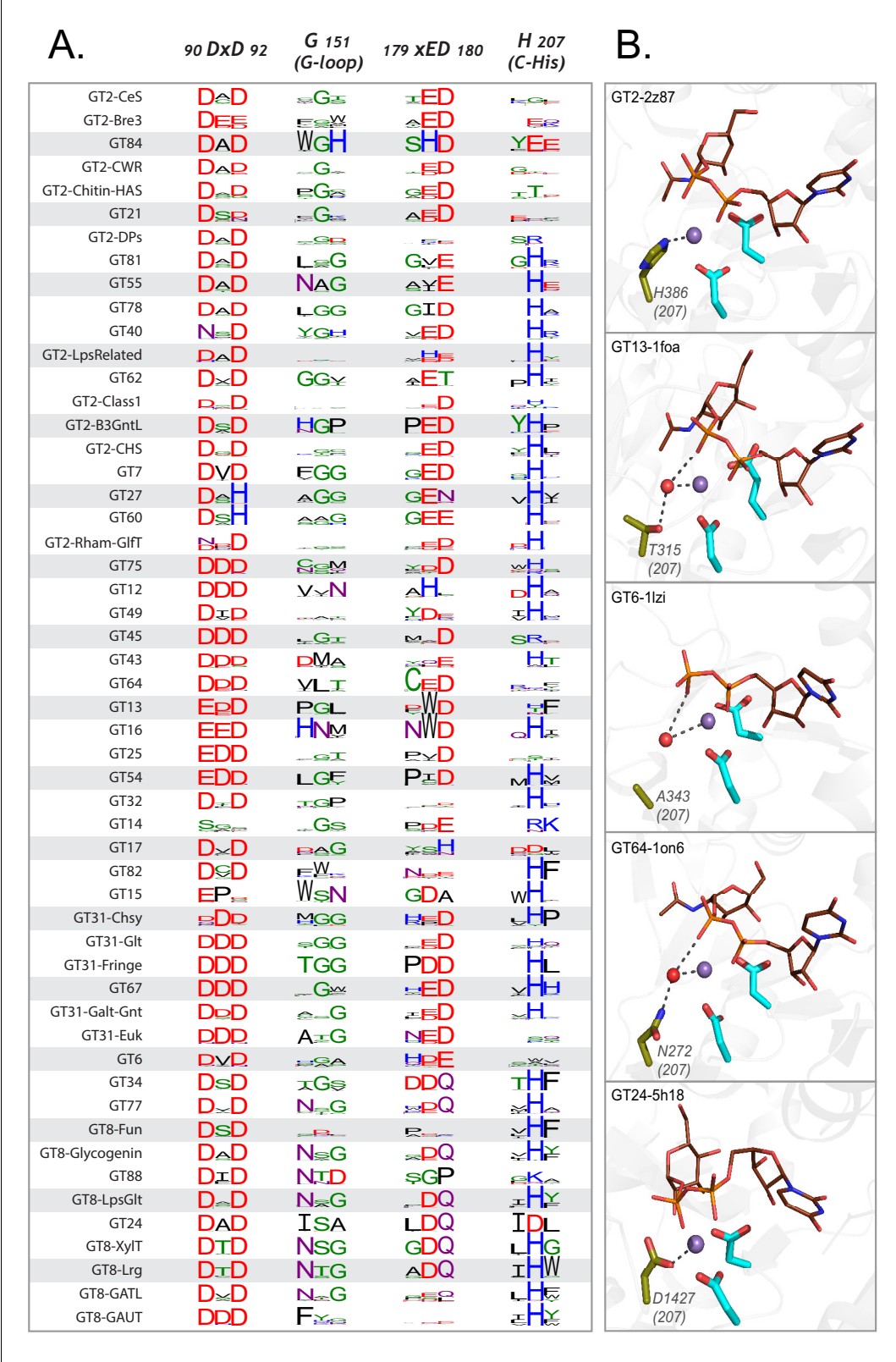

**Figure 4.** Variations in the GT-A conserved core. (A) Weblogo depicting the conservation of active site residues in the common core are shown for each of the GT-A families. Residues are colored based on their physiochemical properties. (B) Variations in the C-His is compensated either using a water molecule (red sphere) or other charged residues (olive sticks) to conserve its interactions. The metal ion is shown as a purple sphere. The donor substrate is shown as brown lines. Interactions between the residues, metal ion and the donor are shown using dotted lines.

families lacking the xED-Asp (GT12, GT14, GT17, GT49 and GT82) are grouped into separate monophyletic clades segregating them from inverting families with the conserved xED-Asp (*Figure 3*). Out of these, only GT14 has representative crystal structures where a glutamate serves as the catalytic base (*Briggs and Hohenester, 2018*). For other inverting families with a non-conserved xED-Asp, residues from other structural regions may serve as a catalytic base. On the other hand, retaining families like GT64 conserve the xED-Asp, yet do not use it as a catalytic base. Thus, there may be multiple ways in which inverting and retaining mechanisms diverge, with one path being mutation of the xED-Asp catalytic base.

One strongly supported clade that includes both inverting and retaining families is clade two that groups inverting GT-A family members that transfer sugars to phosphate acceptors (GT2-DPs) with three retaining GT-A families that also have phosphate-linked acceptors (GT55, GT78 and GT81). This placement is further supported by the observation that these families share structurally equivalent conserved residues in the HV2 region that coordinate the phosphate group of the acceptor. In the GT2-DP subfamily, R117, R131 and S135 (*Figure 5A*) in HV2 coordinate with the acceptor phosphate groups. The conservation of these residues in GT55 and GT81 suggests that they likely perform similar interactions in these latter subclades. Indeed, in the crystal structure of M. tuberculosis GpgS (GT81), HV2 adopts a conformation similar to GT2-DPs and the shared residues G184, R185 and T187 (equivalent to R117, R131 and S135) form similar interactions with the phosphate group of the acceptor (*Figure 3—figure supplement 2*).

Clade five places the inverting GT7 and GT2-CHS with the retaining GT27 and GT60 families (*Figure 3*). This supports the evolution of these families from a close common ancestor through gene

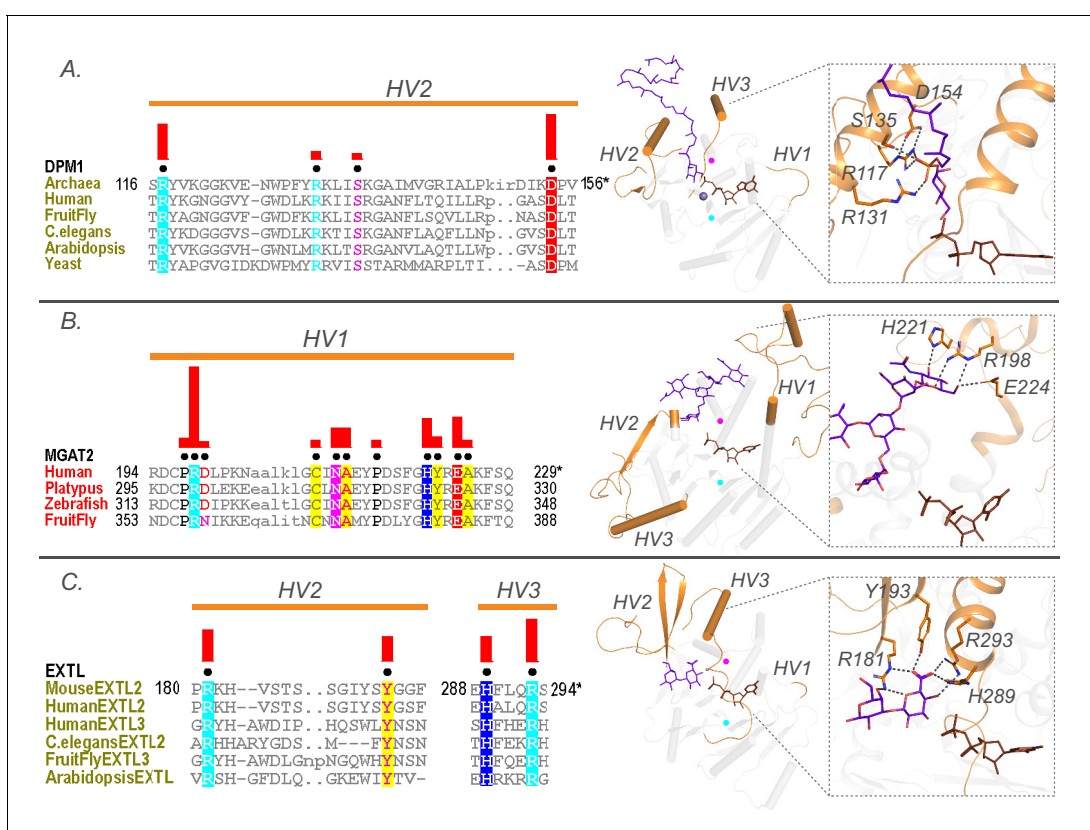

**Figure 5.** Family specific conserved features in the HV regions correlate with acceptor recognition and specificity. Conserved residues in A) HV2 of the DPM1 sequences in the GT2-DP subfamily coordinate the phosphate group of the acceptor. (B) HV1 of GT16 MGAT1 provide acceptor specificity. (C) HV2 and HV3 of EXTL GT64 family (C-terminal GT domain of the multidomain sequences) coordinate the acceptor. Left: Alignments highlighting the constrained residues are shown for each family. The family specific conserved residues are shown using black dots above the alignment. Red bars above these dots indicate the significance of conservation (Higher bar corresponds to more significantly conserved position). Right: Representative pdb structures are shown for each family (GT2-DP:5mm1, GT16:5vcs, GT64:1on8); Donor substrates are colored brown. Acceptors are colored purple. HVs are highlighted in orange. The position of the conserved DxD and xED motif for each structure is shown as cyan and magenta circles respectively.

duplication and divergence, which has been suggested through structural similarities between GT7 and GT27 (*Ramakrishnan and Qasba, 2010b*). After this initial divergence in mechanism within clade 5, the subclades group the β−1,4-GalNAc transferase domains of bacterial and protist chondroitin polymerases (involved in the elongation of glycosaminoglycan chondroitin)(GT2-CHS) with the GT7 family. The GT7 family includes the higher organism counterparts of the β−1,4-GalNAc transferase domains of chondroitin synthases, along with β−1,4-Gal transferases. The close placement of GT60 and GT27 families in this clade is also directly supported by previous literature indicating that these families share a conserved mode of polypeptide Ser/Thr O-glycosylation (*Heise et al., 2009*). Clade five thus consolidates previous independent findings and suggests a shared ancestor, potentially extending the common ancestry of GT2-CHS and GT7 to include GT27 and GT60, with an ancestral divergence in mechanism.

## Variations in the core and hypervariable regions contribute to unique modes of substrate specificity

Analysis of the patterns of conservation and variation in the common core indicates that each residue position within the core has been mutated in some context during the course of evolution, highlighting the tolerance of the GT-A fold to extensive sequence variation. While some of these variations are confined to specific clades or families, such as replacement of DxD motif with DxH motif in GT27 and GT60, other variations are found independently across distal clades (*Figure 4A*). For example, GT14 and prokaryotic members of GT6 that fall on different clades, have independently lost the DxD motif and no longer require a metal ion for activity (*Briggs and Hohenester, 2018*; *Pham et al., 2014*).

The C-His is also lost independently in multiple clades (*Figure 4A*). In order to investigate how the loss of metal binding C-His is compensated, we analyzed the C-His-metal ion interactions across all available crystal structures. Structural alignment of GT-A families lacking the C-His such as GT13, GT6 and GT64 families revealed a water molecule coordinating the metal ion in a manner similar to the C-His sidechain (*Figure 4B*). In other families, such as GT24, we found that the C-His is substituted by an aspartate (D1427), which coordinates with the metal ion similar to C-His (*Figure 4B*, bottom panel). Likewise, the conserved hydrophobic coupling between αF helix and the Rossmann domain is replaced by charged interactions (R388 and E274, respectively) in some retaining GTs such as GT15 and GT55 (*Figure 2—figure supplement 2*). These substitutions point to the ability of GT-As to accommodate changes, even in conserved positions at the core, through compensatory mechanisms.

The HV regions show significant variability across GT-A families and extend from the common core to perform various roles from substrate binding to large conformational changes that position the donor and acceptor substrates for the enzymatic reaction (*Jamaluddin et al., 2007*; *Tsutsui et al., 2013*; *Albesa-Jové et al., 2017*). Mutations within these HV regions, for example, at aligned position 126 in the HV2 region (Y177A,G in 4lw6, GT7), have also been shown to induce a shift in acceptor specificity (*Tsutsui et al., 2013*). Despite significant sequence variability, we find that these HV regions in fact conserve family specific residues that contribute to acceptor specificity. For example, a distinctive arginine (R117) and aspartate (D154) along with R131 and serine S135 within the HV2 of DPM1 (GT2-DP sub-family) contribute to specificity towards a dolichol phosphate acceptor by creating a charged binding pocket for the phosphate group (*Figure 5A*). Likewise, family-specific residues (R198, H221 and E224 in 5vcm) within the HV1 of MGAT2 (GT16) form a unique scaffold for recognizing the terminal GlcNAc of the N-glycan acceptor (*Figure 5B*). Similarly, the C-terminal GT64 domain of the multidomain EXTLs contain specific residues in HV2 (R181 and Y193) and HV3 (H289 and R293) that form a unique binding pocket for the tetrasaccharide linker acceptor used to synthesize glycosaminoglycans (*Figure 5C*). Together these examples illustrate the ability of HVs to evolve family specific motifs to recognize different acceptors.

## ML to predict the donor specificity of GT-A sequences

As discussed above, the conserved catalytic residues dictate the mechanism of sugar transfer and metal binding while the extended HVs use family specific motifs to dictate acceptor specificity. We also find some clade specific features (such as the conserved Lys in clade 9, and QXXRW in clade 1) and G-loop residues involved in donor binding, however, the overall framework that dictates donor

sugar specificity in GTs is largely unknown. Sequence homology alone is insufficient to predict donor specificity because evolutionarily divergent families can bind to common substrates, and sometimes even two closely related sequences bind to different donors (*Figure 6—figure supplement 1*; *Patenaude et al., 2002*). For a subset of GT-B fold families, ML methods have been successfully applied towards predicting substrate specificities (*Yang et al., 2018*). Our global analysis provides a comparative basis to expand such methods and contrast sequences that bind different donors across all GT-A families. To test whether evolutionary features gleaned from this global analysis can be used to better predict donor substrate specificity, we employed a ML framework that learns from the specificity-determining residues of functionally characterized enzymes to predict specificity of understudied sequences. In brief, using an alignment of a well curated set of 713 GT-A sequences (*Figure 6—source data 1*, *Figure 6—figure supplement 2—source data 1*, *Figure 6—figure supplement 2*) with known donor sugars, we derived five amino acid properties (hydrophobicity, polarity, charge, side chain volume and accessible surface area) from each aligned position within the common core. These properties were then used as features to train multiple ML models. Among the seven methods used, the gradient-boosted regression tree (GDBT) model achieved the best prediction performance (accuracy ~90%) based on a 10-fold cross validation (CV) using 239 contributing features (*Figure 6A,B*, *Figure 7—source data 1*). This model adds an ensemble boosting to tree based learners used for predicting GT1 substrate specificities (*Yang et al., 2018*). To further validate the model, we tested its performance on a validation set of 64 sequences that were not used to train the ML model but have known sugar specificities. The GDBT classifier correctly predicted donor substrates for 92% of these sequences, 89% of which were predicted with high confidence (blue rows in *Figure 6—source data 2*).

The GDBT model was then used to predict donor sugars for GT-A domains with unknown specificities from five organisms: *H. sapiens*, *C. elegans*, *D. melanogaster*, *A. thaliana* and *S.cerevisiae* (*Figure 6—source data 2*). Each prediction is associated with a confidence level derived from the probability for each of the six donor classes (Materials and methods). Nearly 77% of the predictions have high and moderate confidence levels and present good candidates for further investigation (*Figure 6D*). The remaining 23% of the predictions are low confidence. This likely reflects their promiscuity for donor preferences, as seen across many GT-As (*Empadinhas et al., 2011*; *Blixt et al., 1999*), or non-catalytic GT-As like C1GALT1C1 (Cosmc) (*Aryal et al., 2012*).

Our predictions assign putative donors for 10 uncharacterized human GT-A domains (*Figure 6—source data 2*). B3GNT9 is predicted to employ UDP-GlcNAc with high confidence like other GT31 β−3-N-acetylglucosaminyltransferases (B3GNTs) in humans (*Togayachi et al., 2010*). The two procollagen galactosyltransferases in humans (COLGALT1 and COLGALT2) are multidomain proteins with two tandem GT-A domains. While their respective C-terminal domains catalyze β-Gal addition to hydroxylysine side-chains in collagen (*Schegg et al., 2009*), our predictions assign a putative GlcNAc and Glc transferase role for their N-terminal GT domains, respectively. More interestingly, GLT8D1, a GT8 GT with an unknown function implicated in neurodegenerative diseases (*Cooper-Knock et al., 2019*), is predicted to have a glucosyltransferase specificity. In other organisms, the GT2 sequences in *A. thaliana* (mostly involved in plant cell wall biosynthesis) are predicted to bind glucose and mannose substrates, the primary components of the plant cell wall (*Figure 6—source data 2*). We also identify a novel N-acetylglucosyltransferase function for a GT25 enzyme in *C. elegans*. These predictions can guide characterization of new GT sequences with unknown functions.

We next sought to identify features that contribute most to substrate (donor) prediction. To do this, we rank ordered the 239 features based on their contribution to predicting a donor subtype using a six way classification (six donors) (Materials and methods). This revealed that the most contributing features of the GDBT model also contribute significantly to at least one specific donor type prediction, thereby enabling new inferences to be drawn between residue properties and donor sugar specificity (*Figure 7A*, *Figure 7—source data 1*). As expected, some of the most contributing features include residues directly involved in substrate binding and catalytic functions such as the Asp within the DxD motif, residues in the G-loop, the catalytic base and the C-His (*Patenaude et al., 2002*; *Empadinhas et al., 2011*; *Gandini et al., 2017*). Additionally, multiple residues from the alpha-C helix (aligned position 65–72; Y217-N224 in 2z87) immediately following HV1 are also identified as key specificity determining residues. The C-helix is positioned close to the donor sugar binding pocket and many residues from this region have been shown to play roles in donor binding (*Gagnon et al., 2018*; *Schuman et al., 2010*; *McArthur and Chen, 2016*). For example,

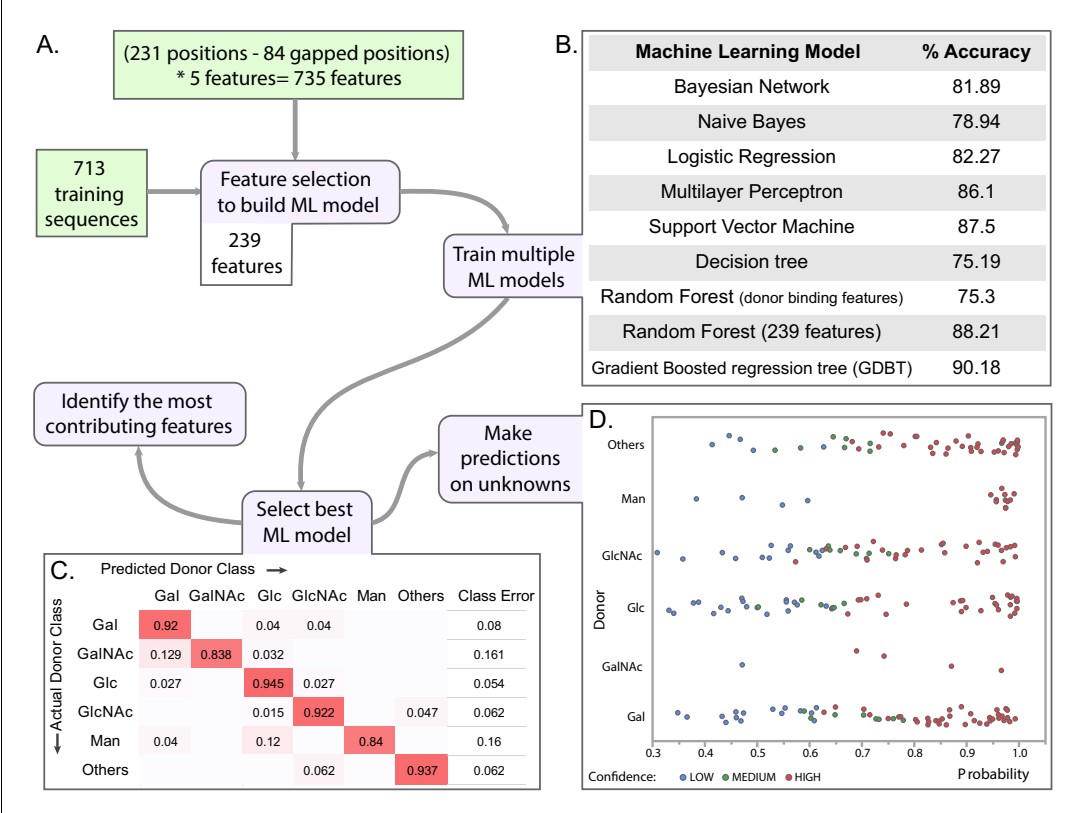

**Figure 6.** Machine learning (ML) approach for predicting donor class. (**A**) Brief pipeline of the ML analysis. Training set input into the pipeline are shown in green boxes. Steps of the ML analysis in purple boxes are associated with different panels of the figure. (**B**) Percent accuracy based on 10-fold cross validation (CV) for each of the trained ML models. (**C**) Confusion matrix from the best model (GDBT using 239 features). (**D**) Scatter plot showing the probability scores assigned for each predicted sequence by the predicted donor type. Colors indicate the confidence level of the prediction based on probability of assignment to a given donor class as well as confidence intervals of the predicted class i.e. difference in probability values between the 1st prediction class and the 2nd prediction class. (**Figure 6—source data 2**).

The online version of this article includes the following source data and figure supplement(s) for figure 6:

**Source data 1.** List of the 713 training dataset sequences used for machine learning.
**Source data 2.** Results for donor prediction using the GDBT ML model for GT-A sequences from five model organisms.
**Figure supplement 1.** Sequence homology-based network of all the experimentally characterized sequences form the GT-A fold families.
**Figure supplement 2.** Distribution of training and prediction datasets used in machine learning.
**Figure supplement 2—source data 1.** Distribution of sequences across different families.

Ramakrishnan et. al. showed that mutation of a single residue at position 67 in bovine $\beta-1,4$-galactosyltransferase T1 (R228K in 1o0r, GT7) resulted in relocation of the catalytic base and a change of donor specificity from Gal to Glc (**Figure 7B**; **Ramakrishnan et al., 2005**; **Hancock et al., 2006**). Our analysis identifies volume, polarity and accessible surface area of the residue at position 67 as an important contributor to donor specificity (**Figure 7**). In addition, our analysis identifies residue volume at position 149 as an important determinant of Gal specificity. Consistent with this observation, mutation of Y289 (position 149 in the consensus sequence) by a leucine broadens the specificity from Gal to GalNAc by creating additional space for accommodating the N-acetyl moiety (**Hancock et al., 2006**; **Ramakrishnan and Qasba, 2002**).

While some of the highly ranked features are directly involved in donor binding, many others (such as aligned position 77, 88, 155 and 159, green sticks in **Figure 7B,D**) are distal from the donor binding site and are not directly involved in donor binding. An example of allostery has been observed in the human GT6 blood ABO $\alpha-1,3$-galactosyltransfearse where mutation of a proline at position 117 (P234S in 5c4c) results in an alternative conformation of a methionine at position 150 (M266 in 5c4c) allowing for the accommodation of GalNAc instead of Gal (**Hancock et al., 2006**;

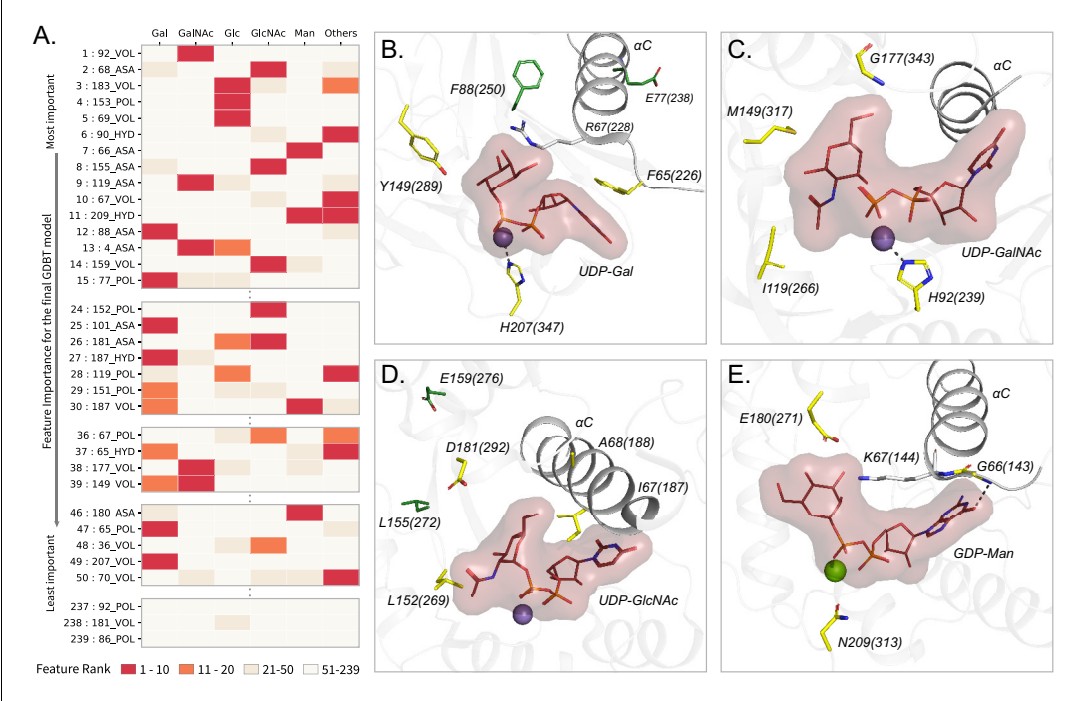

**Figure 7.** Top Contributing features from the GDBT model associated with sugar donor specificity. (**A**) Heatmap showing the contributions of representative features. Features are ordered based on their importance for the final GDBT model along the vertical axis. The heatmap colors indicate how important each feature is for a given sugar donor type with red indicating ranks 1–10 (highly important) (M). (**B–E**) Contributing features important for individual donor types are mapped onto representative structures. The amino acids at the feature positions are shown in yellow sticks and labelled. Feature positions distal from the donor binding site are shown in green sticks. Labels include the amino acid code, aligned residue position and the amino acid position in the crystal structure within parenthesis. Donor substrate with the sugar is shown in lines with surface bounds. Divalent metal ions are shown as spheres. The αC helix is shown. (**B**) Gal features mapped to a bovine β−1,4 Gal transferase (PDB ID: 1o0r). (**C**) GalNAc features mapped to a human UDP-GalNAc: polypeptide alpha-N-acetylgalactosaminyltransferase (PDB ID: 2d7i). (**D**) GlcNAc features mapped to a rabbit N-acetylglucosaminyltransferase I (PDB ID: 1foa). (**E**) Man features mapped to a bacterial Mannosyl-3-Phosphoglycerate Synthase (PDB ID: 2wvl). The online version of this article includes the following source data for figure 7:

**Source data 1.** Feature Importance comparison for the full GDBT model with its importance for each sugar donor type.

---

*Marcus et al., 2003*). Further, a Random forest model trained using features from only the donor binding residues performs with an accuracy of only 75%, indicating the importance of features other than those directly involved in donor binding. Thus, despite only a few residues being directly involved in donor interactions, additional contributions to donor specificity come from residues more distal from the active site. Contributions from these peripheral secondary shell features surrounding the donor binding site (*Figure 7B–E*) highlight the potential role of higher order (allosteric) interactions in determining donor substrate specificity.

## Discussion

Prior studies on the evolution of GTs have generally focused either on distinct GT subfamilies or biosynthetic pathways with additional structural classifications of GTs into one of three distinct protein fold superfamilies (*Moremen and Haltiwanger, 2019*; *Taujale and Yin, 2015*; *Lombard, 2016*). In our present work we focused on the analysis of the largest of the GT superfamilies, those that comprise a GT-A protein fold characterized by an extended Rossmann domain with associated conserved helical segments. These enzymes generally employ the Rossmann domain for nucleotide sugar donor interactions and extended loop regions for acceptor glycan interactions (*Moremen and Haltiwanger, 2019*). Using an unbiased profile search strategy, we assembled a total of over 600,000 GT-A fold related sequences from all domains of life for deep evolutionary analysis. To support this profile-based assembly, we leveraged structural alignments on GT-A fold enzymes in PDB

and secondary structure predictions when no crystal structures were available. The resulting alignment allowed the definition of a common structural core shared among the diverse GT-A fold enzymes and defined positions where hypervariable loop insertions were elaborated to provide additional functional diversification (*Figure 2*). In cases where data was available for enzyme-acceptor complexes these latter loop insertions generally contribute to unique, family specific acceptor interactions. Thus, a structural framework is presented for GT-A fold enzyme evolution. Since the common core is present across all kingdoms of life, it presumably represents the minimal ancestral structural unit for GT-A fold catalytic function by defining donor substrate interactions and minimal elements for acceptor recognition and catalysis. In fact, we find several archaeal and bacterial sequences that closely resemble this common core consensus sequence (*Supplementary file 3*). Based on our studies, we propose a progressive diversification of GT function through evolution of donor specificity by accumulation of mutations in the common core region and divergence in acceptor recognition through expansion of the hypervariable loop regions. Consistent with this view, we find conserved family-specific motifs within the hypervariable regions that confer unique acceptor specificities in various families. These expansions likely contributed to the evolution of new GT functions and catalyzed new glycan diversification observed in all domains of life.

A surprising finding from our studies is the dispersion of inverting and retaining catalytic mechanisms among families in the GT-A fold evolutionary tree (*Figure 3*). Recent models indicate that distinctions between inverting and retaining catalytic mechanisms arise from differences in the angle of nucleophilic attack by the acceptor toward the anomeric center of the donor sugar (*Moremen and Haltiwanger, 2019*). Inverting mechanisms require an in-line attack and direct displacement by the nucleophile relative to the departing nucleotide diphosphate of the sugar donor and a conserved placement of the xED-Asp carboxyl group as catalytic base at the beginning of the αF helix. In contrast, retaining enzymes generally alter the angle of nucleophilic attack by the acceptor, use a donor phosphate oxygen as catalytic base, and employ a dissociative mechanism for sugar transfer (*Moremen and Haltiwanger, 2019*). The fundamental differences in these catalytic strategies would suggest an early divergence of enzymes employing these respective mechanisms. However, the GT-A fold phylogenetic tree strongly suggests that inverting and retaining mechanisms evolved independently at multiple points in the evolution of GT-A families (*Figure 3*). Since the main difference in these mechanisms is the change in position of the nucleophilic hydroxyl and catalytic base, substitutions at the catalytic base may have served as a catalyzing event in switching between mechanisms. The xED-Asp carboxyl group is highly conserved in the inverting enzymes and is appropriately placed for acceptor deprotonation. Variants of this motif either lack the residue entirely, as seen in many retaining enzymes, or use compensatory modes to accommodate changes at this position, as seen for the inverting enzymes in GT43, GT2-DPs, and GT2-LPSRelated. In fact, in each of the latter cases the respective inverting GT family is clustered with closely related GT families employing a retaining catalytic mechanism. Thus, inverting enzyme variants that accommodate changes to the xED motif group may represent examples of transitional phases in evolution between inverting and retaining catalytic mechanisms. Other inverting enzymes harboring variants in the xED motif segregate into separate clades and could represent outlier families that have developed alternative ways to compensate for the loss of xED-Asp. This ability to evolve distinct catalytic strategies, in some cases through presumed convergent evolution, could allow each family to evolve independent capabilities for donor and acceptor interactions as well as for anomeric linkage of sugar transfer, while retaining other essential aspects of protein structural integrity through the use of a conserved and stable Rossmann fold core.

In an effort to define the sequence constraints for the respective catalytic mechanisms we also employed a ML framework for prediction of the mechanism for unknown sequences and were able to assign the donor sugar nucleotide for a test set of enzymes with high accuracy. Our model expands on the approaches used in previous ML efforts focused on the GT1 family of GT-B fold GTs (*Yang et al., 2018*). The phylogenetic and comparative framework presented here enables expansion of such models across all GT-A fold families with improved prediction accuracies. As additional functional data on GTs become available, the proposed ML framework can be extended to predict acceptor specificity and catalytic mechanisms, as described for the GT1 family. Surprisingly, the contributing features for accurate donor prediction include residues involved in donor binding as well as positions that are distal to the active site that likely contribute through secondary shell effects or allosteric interactions. Due to their indirect involvement, such positions are generally difficult to

pinpoint using structural studies alone emphasizing the need for complementary ML-based approaches in investigating GT functional specialization.

Numerous additional insights into GT function were also revealed through inspection of the aligned sequences and the phylogenetic tree. For example, the clustering of mammalian N-glycan GlcNAc branching enzymes (MGAT1 (GT13), MGAT2 (GT16), and MGAT4 (GT54)) in the same clade suggests a common origin for these enzymes, while placement of MGAT3 (GT17) in a separate clade could point to its unique role in adding a bisecting GlcNAc to the N-glycan core thereby regulating N-glycan extension (*Ikeda et al., 2014*). In contrast, MGAT5 (GT18) involved in N-glycan β1,6-GlcNAc branching is a GT-B fold enzyme with a clearly distinct evolutionary origin. While most clades are well resolved, bootstrap support values for nodes at the base of the tree are low and need to be interpreted with caution. This low resolution results from high divergence between families and possibly other events like horizontal gene transfer and convergent evolution. However, trees generated using alternative strategies support the overall topology (*Figure 3—figure supplement 3*) and clades are congruent with clusters obtained using an orthogonal Bayesian classification scheme, which adds confidence to the phylogeny (*Figure 3—source data 1*).

For some GT-A fold enzymes variations in the catalytic site can also be accommodated by other compensatory changes. An example is the use of the C-His motif for coordination of the divalent cation in most GT-A fold enzymes in contrast with enzyme variants that employ water molecules to compensate for the loss of this residue (*Figure 4B*). Similarly, some inverting GTs dispense with the use of the divalent cation and the DxD motif and substitute interactions with the sugar donor through use of basic side chains (e.g. GT14). A further extreme is the duplication, divergence and pseudogenization within the GT31 family. Human C1GALT1C1 (GT31, COSMC) shares a high sequence similarity to another GT31 member, C1GALT1 (T-synthase), yet COSMC has lost both the DxD and the xED motifs and has no catalytic activity. Instead, COSMC acts as an important scaffold and chaperone for the proper assembly and catalytic function of T-synthase (*Aryal et al., 2012*). The ability of GT-As to harbor such structural variations that allow them to develop new functions make them well-suited to evolve rapidly and facilitate the synthesis of a diverse repertoire of glycans across all living organisms.

Our unbiased, top-down sequence-based analysis suggests new and unanticipated evolutionary relationships among the GT-A fold enzymes. Prior suggestions of such relationships have been inferred by the clustering of GT sequences into families in the CAZy database. However, the CAZy database of GT sequences does not provide access to the broader sequence relationships among the GT-A fold enzymes or how a general model of a core conserved GT-A fold scaffold can serve as a progenitor catalytic platform for binding sugar donors and facilitating glycan extension. The sequence assembly, phylogenetic tree, and placement within the framework of known GT-A fold structures in the present studies provide key insights into conserved elements of the hydrophobic core, linkage to the DxD motif for cation and sugar donor interactions, and the conserved αF helix harboring the xED catalytic base. Additional hypervariable extensions at defined positions from this conserved core were then progressively recruited to confer unique modes of acceptor interactions to develop new specificities and evolve new functions. Thus, the core of the protein scaffold can be maintained to facilitate protein stability while rapid evolution of the hypervariable loops can develop new glycan synthetic functionalities through presentation of novel acceptors to the catalytic site. Variation in the location of the acceptor hydroxyl nucleophile relative to the donor sugar anomeric center presents the opportunity for distinctions in catalytic mechanism and anomeric outcome for sugar transfer. The result is a rapidly evolving set of GT enzymatic templates as the biosynthetic machinery for diverse glycan extension on cell surface and secreted glycoproteins and glycolipids. In such contexts the resulting glycoconjugates confer potential functional selective advantages at the cell surface, but also act as ligands and pathogen entry points for negative evolutionary pressure. These positive and negative selective pressures which force organisms to constantly adapt to an ever-changing environment is known as the Red Queen Hypothesis. These red queen effects on glycan synthesis have led to the remarkable diversity in GT enzymes and their resulting glycan structural products. We anticipate that the sequence and structural principles that drive GT-A fold evolution will also likely extend to GT-B and GT-C fold enzymes and represent a common theme for the elaboration of diverse glycan structures in all domains of life.

# Materials and methods

**Key resources table**

| Reagent type (species) or resource | Designation | Source or reference | Identifiers | Additional information |
|---|---|---|---|---|
| Software, algorithm | CAZy database | doi: 10.1093/nar/gkt1178 | CAZy- Carbohydrate Active Enzyme, RRID:SCR_012909 | |
| Software, algorithm | mapgaps | doi: 10.1093/bioinformatics/btp342 | | |
| Software, algorithm | omcBPPS | doi: 10.1089/cmb.2013.0099 | | |
| Software, algorithm | GT-A family classification and sequences | This paper | doi: 10.5061/dryad.v15dv41sh | |
| Software, algorithm | MAFFT v7.3 | doi: 10.1093/molbev/mst010 | MAFFT, RRID:SCR_011811 | |
| Software, algorithm | Expresso from the t-coffee suite | doi: 10.1093/nar/gkl092 | T-Coffee, RRID:SCR_011818 | |
| Software, algorithm | IQTree v1.6.1 | doi: 10.1093/molbev/msu300 | | |
| Software, algorithm | PyMOL v2.0.6 | Schrödinger | PyMOL, RRID:SCR_000305 | |
| Software, algorithm | Python v3 with package scikitlearn | *Pedregosa, 2011* | scikit-learn, RRID:SCR_002577 | |
| Software, algorithm | R package 'randomForest' | *Liaw and Wiener, 2002* | RandomForest Package in R, RRID:SCR_015718 | |
| Software, algorithm | WEKA version 3.8.3 | *Witten et al., 2016* | Weka, RRID:SCR_001214 | |

## Generation of GT-A profiles and alignment

### Building the GT-A profiles

Multiple alignments for 34 CAZy GT-A families, as determined based on literature (*Breton et al., 2012*; *Lombard et al., 2014*; *Liu and Mushegian, 2003*; *Breton et al., 2006*), were collected from the Conserved Domain Database (CDD) (*Marchler-Bauer et al., 2017*) or were manually built using MAFFT v7.3 (*Katoh and Standley, 2013*) from sequences curated at the CAZy database (*Supplementary file 1*). Multiple separate alignments were generated for large families such as GT2 and GT8 to capture the diversity within these families. These alignments made up the seed profiles for the GT-A families. These seed profiles were then multiply aligned using the mapgaps scheme (*Neuwald, 2009*) guided by a structure based sequence alignments of all available pdb structures using Expresso (*Armougom et al., 2006*) and MAFFT to generate the GT-A profiles. Representative pdb structures described in this study are listed and cited in *Supplementary file 2*. Alignments for families with no representative crystal structures were guided using secondary structure predictions performed using PCI-SS (*Green et al., 2009*). Finally, the alignment of secondary structures and conserved motifs were manually examined and corrected, where necessary. Very divergent GT-A families, such as GT29 and GT42 sialyltransferases, lack nearly all canonical GT-A motifs and do not align well with other GT-A families. Thus, they are noted as atypical GT-A fold families and not included in this analysis.

### Sequence alignment and defining the GT-A common core

The GT-A profiles were then used for a sequence similarity search using mapgaps to identify and align ~600,000 GT-A domain sequences from the NCBI non redundant database. This alignment was filtered for fragmentary sequences and false hits. This filtered alignment was then used to define the boundaries of the GT-A common core that extends from the first beta sheet of the Rossmann fold to a C-terminal helix with family specific motifs. This conserved alignment spanned 231 aligned

positions. Sequences with multiple GT-A domains (like the GT8 and GT49 LARGE domains) or other accessory domains (like the GT27 and lectin domains) were separated into individual catalytic GT-A domains and treated separately throughout the analyses.

## Structural alignment of Rossmann fold proteins

A select representative set of structures were collected from all Rossmann-fold containing protein domains using the SCOP database (*Andreeva et al., 2014*). mTM-align (*Dong et al., 2018*) was used to align these structures with a subset of GT-A structures (*Figure 2—figure supplement 1*).

## Bayesian statistical analyses

A representative subset of 24,650 GT-A sequences were generated from the ~600,000 putative GT-A sequences by using a family-wise sequence similarity filtering (only keep <70% similar sequences;<50% for GT2 and GT8 families). This sequence set was then used to apply the Optimal multiple-category Bayesian Partitioning with Pattern Selection (omcBPPS) scheme (*Neuwald, 2014*). omcBPPS identifies patterns of column-wise amino acid conservation and variation in the multiple sequence alignment. The resulting family specific positions were then used as statistical measures to classify the GT-As into 99 unique sets that correspond to the 53 families described in this study (*Figure 3—source data 1*). omcBPPS also identified aligned positions that are conserved across all GT-A fold families. This revealed the 20 conserved positions within the core component, that were also verified by calculating conservation scores using the Jensen-Shannon divergence score as described and implemented by *Capra and Singh (2007)* (used in *Figure 2A*).

## Phylogenetic analysis

### Selection of sequences for phylogenetic analysis

A smaller subset of 993 sequences were used for phylogenetic reconstruction of the GT-A families (*Figure 3—source data 2*). This set includes all the identified GT-A sequences from five model organisms: *H. sapiens* (human), *C. elegans* (worm), *D. melanogaster* (fly), *A. thaliana* (dicot plant) and *S. cerevisiae* (yeast) along with select sequences representing the diverse taxonomic group in each family. These representative sequences were selected by finding the union of top hits for every taxonomic group present within each of the 99 sets and the seed alignments for the 34 CAZy GT-A families. This selection criteria maximized the phylogenetic and taxonomic diversity while keeping the number of sequences to a minimum.

### Details of the phylogenetic inference

The alignment for these 993 sequences was trimmed to remove the insert positions and keep only the 231 aligned positions described above. This trimmed alignment was used to build a phylogenetic consensus tree using IQTree v1.6.1 (*Nguyen et al., 2015*) with the following options: -nt AUTO -st AA -m MFP+MERGE -alrt 1000 -bb 1000 -wbt -nm 1000 -bnni. This implements ModelFinder (*Kalyaanamoorthy et al., 2017*) to select the best fit model based on Bayesian Information Criterion (BIC). Clade support for this tree was evaluated using bootstrapping which reports support values based on the number of times the same clade was observed on 1000 trees built using resampled alignment. Clades with bootstrap support values over 90% are well supported while values over 75% are moderately supported. Clades with bootstrap values less than 50% are considered unresolved in our analysis.

### Orthogonal support for the phylogenetic tree

Further support for the phylogenetic tree was collected by comparing its topology to trees generated using orthogonal methods like Hidden Markov Model (HMM) distances and structural similarities, that have been used in previous studies (*Huo et al., 2017*; *Hashimoto et al., 2010*; *Figure 3—figure supplement 3*). The HMM-distance based phylogenetic tree was built using pHMM-Tree (*Huo et al., 2017*). Briefly, hmm profiles were built for each of the 53 sub-families identified in our analyses. Pairwise distances between these profiles were calculated and the resulting distance matrix was used to build a neighbor joining tree. All trees were visualized using the interactive Tree of Life (iTOL) online tool (*Letunic and Bork, 2019*). For the structural similarity based clustering, pairwise root mean square distances (RMSD) were calculated for 50 unique representative GT-A structures

using the cealign algorithm in PyMol v2.0.6 (*Schrödinger, LLC, 2017*) to build a distance matrix. Only the defined GT-A catalytic domain spanning the 231 aligned positions along with insertions were used for the RMSD calculations. This RMSD matrix was then used for clustering using the 'ward' method in python which resulted in a structural distance based hierarchical clustering of the pdb structures. The hierarchical topology obtained from the HMM distance-based method and the RMSD distance based clustering were then compared to the tree topology in *Figure 3* (*Figure 3— figure supplement 3*). Non-overlapping connections show consistently placed families.

## Defining the GT-A families and sub-families

The GT-A sequences were first classified into pattern-based groups using omcBPPS. Based on the placement of representative sequences from these groups in the phylogenetic tree, they were merged into GT-A families and sub-families. The correspondence between the 53 GT-A families and subfamilies with the 99 pattern-based groups are provided in *Figure 3—source data 1*. Sequences from some families did not form any distinct pattern-based groups due to either a low number of sequences for a statistically significant grouping (GT78) or a lack of distinguishing patterns within the aligned positions (GT25, GT88). Representative sequences for these families were collected from the seed alignments for these families as described above. We also identified the N-terminal GT2 domain of the multidomain chondroitin polymerase structure from *E. coli* (Pdb Id: 2z87) as the proto-typic GT-A structure to use as a comparative basis for structural analyses. This sequence was selected based on the lowest E-value and highest similarity score of a BLAST search of all pdb structures against the GT-A consensus sequences. Weblogos for the conserved active site residues were derived for each GT-A subfamily using Weblogo 3.6.0 (*Crooks et al., 2004*).

## ML analysis

### Gathering the training and validation dataset

In order to train an ML model for GT-A donor substrate prediction, we first curated a training dataset by mining the 'characterized' tab of the CAZy GT database and the UniProt database (*UniProt Consortium, 2019*) to find 713 GT-A domain sequences with known donor sugars. The donor sugar information for these sequences were extracted from their assigned protein names. Based on the availability of training sequences, six major donor type classes were defined: Glc, GlcNAc, Gal, GalNAc, Man, and 'Others' with each class having more than 70 sequences in the training dataset. The 'Others' category merged the least represented donor types with less than 50 training sequences each (Ara, Fuc, GalF, GlcA, ManNAc, Rham, and Xyl). An alignment of the 713 sequences was generated and then used to derive five amino acid properties (charge, polarity, hydrophobicity, average accessible surface area, and side chain volume) (*Kawashima et al., 1999*) for each aligned position. These properties were used as features for ML. We first removed highly gapped positions (>15% gaps) and implemented correlation-based feature selection (CFS) (*Hall, 1999*) with 5-fold CV by using WEKA version 3.8.3 (67) under default settings to select 239 informative features for building multiple multiclass classification models. In addition, we also curated 64 GT-A sequences with known donor sugars for five model organisms (*H. sapiens*, *C. elegans*, *D. melanogaster*, *A. thaliana* and *S.cerevisiae*). These sequences were not used to train the ML model but set aside to be used as validation dataset to test the performance of the model (*Figure 6—source data 2*).

### ML model training

We first trained random forest models by using an R package 'randomForest' (*Liaw and Wiener, 2002*) with limited number of trees (ntree = 300) and limited maximum number of terminal nodes (maxnodes = 100) to avoid unrestricted tree expansion and potential overfitting. Two separate models were trained where the first one was trained with the larger set of 239 features and the second model was provided only 25 features coming from the donor binding residues. We used the GradientBoostingClassifier function of the sklearn package in python (*Pedregosa, 2011*) to train the gradient boost regression tree (GDBT) model on the 239 features. This model was trained with the following parameters: learning_rate = 0.1, n_estimators = 1600, min_samples_split = 25, min_samples_leaf = 7, max_depth = 4, max_features = 18, subsample = 0.75 and random_state = 10. These parameters were chosen based on a grid search to fine tune the trade-off between the complexity

of the model and the metrics on the testing data, thus ensuring meaningful predictions and avoiding overfitting. The importance of each feature used in the GDBT model was measured based on the relative rank of the features in the decision nodes of a tree. To compare the performance of these models, we also trained Support Vector Machine (SVM), multilayer perceptron, Bayesian network, logistic regression, naive Bayes classifier, and decision tree models by using WEKA with 10-fold CV under default settings. 10-fold CV evaluates the ML models by iteratively training on 90% of the data selected at random and testing the prediction on the unseen 10% of the data. This is repeated 10 times and the results on the testing dataset are summarized into an accuracy measure. The GDBT model trained with 239 features had the highest accuracy and overall performance and thus was selected as the model of choice for predicting donor sugar substrates for GT-A enzymes.

## Evaluating the confidence of predictions

Confidence scores were assigned for each prediction based on the probability for each of the six donor classes. The class with the highest probability represents the predicted donor sugar. As such, larger differences in probability between the first and second predicted class result in more reliable predictions. To interpret this difference in score easily, we derive a three-category confidence level. If the probability for the first class is more than four times the probability of the second predicted class, then it is considered a high confidence prediction. If the difference is less than four times but more than double the probability by random chance (2*1/6 for a six class classification), it is considered a moderate confidence prediction. If it is neither, then it is a low confidence prediction (*Figure 6—source data 2*).

## Determining feature contributions for each donor sugar specificity

Feature importance for the GDBT model was first assessed using the relative rank of the features in the decision tree. Then, six separate GDBT models were trained as binary classifiers (Gal Vs everything else, Glc Vs everything else and so on for the six donor types). For each of these six classifiers, the features were rank ordered in the same way by assessing their rank in the decision tree nodes. This provided the contributions of each of the 239 features toward a specific donor specificity which was then compared to its rank in the full GDBT model (*Figure 7—source data 1*).

## Construction of the sequence similarity network

A sequence similarity network was generated to evaluate the general predictability of a donor substrate based on sequence homology alone. Using an Edge-Weighted Spring Embedded Layout (*Kamada and Kawai, 1989*) in Cytoscape (*Shannon et al., 2003*) with the same sequence dataset as the ML data, we produced a series of homologous networks constrained by an E-value cut-off of 0.05.

## Selection of ancient archaeal and bacterial GT-A domain sequences in *Supplementary file 3*

The ancient archaeal and bacterial GT-A domain sequences listed in *Supplementary file 3* represent sequences with the minimal GT-A domains (no long inserts, no additional domains) in prokaryotic organisms. These sequences could represent the most ancient progenitors of the GT-A fold families. These were selected based on closest homology to a consensus sequence derived from the GT-A profile alignments. First, a BLAST (*Altschul et al., 1990*) search was conducted with the consensus sequence on the NCBI nr database. The 500 hits with an e-value better than 1e-10 were selected. These hits were then aligned to the GT-A profiles using mapgaps. Using this alignment, these sequences were further filtered to include hits that had a) more than 140 aligned positions to remove fragmentary hits, b) less than 20 insert positions throughout the GT-A domain alignment, c) no more than 100 amino acids in the N-terminal region and d) no more than 200 amino acids in the C-terminal region.

## Acknowledgements

Funding for NK and KWM from NIH R01GM130915 is acknowledged. RT was also supported by T32 GM107004.

## Additional information

### Funding

| Funder | Grant reference number | Author |
|---|---|---|
| National Institutes of Health | R01 GM130915 | Kelley W Moremen<br>Natarajan Kannan |
| National Institutes of Health | T32 GM107004 | Rahil Taujale |

The funders had no role in study design, data collection and interpretation, or the decision to submit the work for publication.

### Author contributions

Rahil Taujale, Conceptualization, Data curation, Formal analysis, Validation, Investigation, Visualization, Methodology, Writing - original draft, Writing - review and editing; Aarya Venkat, Formal analysis, Investigation, Visualization, Methodology, Writing - original draft, Writing - review and editing; Liang-Chin Huang, Formal analysis, Validation, Methodology, Writing - review and editing; Zhongliang Zhou, Software, Formal analysis, Validation, Investigation, Writing - review and editing; Wayland Yeung, Visualization, Methodology, Writing - review and editing; Khaled M Rasheed, Validation, Methodology, Writing - review and editing; Sheng Li, Supervision, Validation, Investigation, Methodology, Writing - review and editing; Arthur S Edison, Conceptualization, Supervision, Writing - review and editing; Kelley W Moremen, Conceptualization, Supervision, Funding acquisition, Investigation, Methodology, Writing - review and editing; Natarajan Kannan, Conceptualization, Resources, Software, Supervision, Funding acquisition, Methodology, Writing - original draft, Writing - review and editing

### Author ORCIDs

Rahil Taujale (iD) https://orcid.org/0000-0003-1292-1619
Aarya Venkat (iD) https://orcid.org/0000-0002-8793-4097
Arthur S Edison (iD) http://orcid.org/0000-0002-5686-2350
Natarajan Kannan (iD) https://orcid.org/0000-0002-2833-8375

### Decision letter and Author response

Decision letter https://doi.org/10.7554/eLife.54532.sa1
Author response https://doi.org/10.7554/eLife.54532.sa2

## Additional files

### Supplementary files

• Supplementary file 1. The CAZy GT-A families included in the analysis. The 'Alignment source' column includes the CDD alignment identifiers used to build the seed profiles for a given GT-A family. If a suitable CDD profile was not available, the seed profiles were built by manually selecting and aligning representative sequences.

• Supplementary file 2. Mapping of the 231 aligned positions to representative crystal structures available for all GT-A fold families. The 'Position Description' column includes the labels for conserved motifs and hypervariable regions for the aligned positions. GT family, PDB IDs and reference to PubMed IDs are indicated at the header columns.

• Supplementary file 3. The ancient archaeal and bacterial sequences that most closely resemble the GT-A core consensus. These sequences were collected by running a BLAST search against a single consensus sequence generated from the seed profiles of all GT-A families. The hits were further filtered to keep sequences that only had the minimal GT-A core (Materials and methods).

• Transparent reporting form

## Data availability

All the data generated during the study are summarized and provided in the manuscript and supporting files. Source files have been provided for Figures 1, 3, 6 and 7. Additionally, all the sequences curated during this study have been deposited to Dryad (https://doi.org/10.5061/dryad.v15dv41sh).

The following dataset was generated:

| Author(s) | Year | Dataset title | Dataset URL | Database and Identifier |
|---|---|---|---|---|
| Taujale R, Venkat A, Huang LC, Yeung W, Rasheed K, Edison AS, Moremen KW, Kannan N | 2020 | Deep evolutionary analysis reveals the design principles of fold A glycosyltransferases | https://doi.org/10.5061/dryad.v15dv41sh | Dryad Digital Repository, 10.5061/dryad.v15dv41sh |

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
