## [Decision Letter]

**Acceptance summary:**

The basis for the diverse donor and acceptor substrate specificity of glycosyltransferases was probed by global sequence analysis. Deep mining of 600,000 sequences of glycosyltransferase members of the GT-A fold family defined a minimal common core and hypervariable regions for these functionally diverse enzymes. Variations in these regions contribute to diversification of substrate specificity. A phylogenetic tree was generated from a representative set of GT-A sequences that showed retaining and inverting mechanisms evolved independently multiple times. The analysis revealed new clustering of GT-A sub families and clades, previously undescribed by CAZy. Machine learning was used to predict donor sugar specificity addressing a major problem for glycosyltransferases given the low numbers of biochemically and functionally characterized GTs. The study design goes beyond a simple evolutionary mapping, documenting supporting evidence for their conclusions with selected enzymes and using data to create a tool for predicting substrate use.

**Decision letter after peer review:**

Thank you for submitting your article "Deep evolutionary analysis reveals the design principles of fold A glycosyltransferases" for consideration by *eLife*. Your article has been reviewed by three peer reviewers, including Monica Palcic as the Reviewing Editor and Reviewer #1, and the evaluation has been overseen by Michael Marletta as the Senior Editor. The following individuals involved in review of your submission have agreed to reveal their identity: Karen Colley (Reviewer #2); Stephen Withers (Reviewer #3).

The reviewers have discussed the reviews with one another and the Reviewing Editor has drafted this decision to help you prepare a revised submission.

Summary:

The basis for the diverse donor and acceptor substrate specificity of glycosyltransferases (GTs) was probed by global sequence analysis. Deep mining of 600,000 sequences of glycosyltransferase members of the GT-A fold family defined a minimal common core and hypervariable regions for these functionally diverse enzymes. Variations in these regions contribute to diversification of substrate specificity. A phylogenetic tree was generated from a representative set of GT-A sequences that showed retaining and inverting mechanisms evolved independently multiple times. The analysis revealed new clustering of GT-A sub families and clades, previously undescribed by CAZy. Machine learning was used to predict donor sugar specificity addressing a major problem for glycosyltransferases given the low numbers of biochemically and functionally characterized GTs.

The major conclusions and the design of the study are of sufficient interest for publication in *eLife*. The study design goes beyond a simple evolutionary mapping, documenting supporting evidence for their conclusions with selected enzymes and using data to create a tool for predicting substrate use.

The manuscript is well written requiring editorial correction as noted in the reviewers' minor comments. It would be nice to include figures showing retaining and inverting glycosyltransferase mechanisms and examples of the three glycosyltransferase folds with their abundances for general *eLife* readers. Explain the Red Queen effect for readers unfamiliar with the analogy. Prior analysis of Arabidoposis family 1 members comprised of the GT-B fold should also be discussed including comparison of the machine learning approaches. Analysis of experiments on the directed evolution of GT-A members would provide additional support for the suggestion that alterations in substrate specificity arise from hypervariable loop mutations. The distribution of the 713 members of the test set within the phylogenetic tree is also of interest.

*Reviewer #1:*

This is a much needed analysis of the entire GT-A fold family of glycosyltransferases (GTs) focusing on their conserved structural features and evolutionary relationships within the framework of their CAZy defined families. Key findings are the identification of conserved secondary structures and hypervariable regions characteristic of the GT-A family including a common core with hydrophobic and active site DXD/xED metal binding residues, a G-loop and C-His. The core alignment was used to generate a phylogenetic tree for a subset of representative sequences.

The phylogenetic tree defines sub-families within large CAZy GT families (GT2, GT8 and GT31) and groups GT families and sub-families into clades. This is a significant achievement given that the lack of sub-groups within the large GT families and identification of clades is a limitation of the CAZy database.

An intriguing feature of the phylogenetic tree was the location of mechanistically distinct inverting and retaining enzymes throughout the identified clades. This suggests that retaining and inverting enzymes appeared independently multiple times through GT-A evolution.

The prediction of donor sugar specificity addresses a major problem for glycosyltransferases given the low numbers of biochemically and functionally characterized GTs. This was based on a machine learning of donor types, however, this prediction is for the sugar transferred from the donor and not the nucleotide. For example, glucose is transferred by different GTs from UDP-Glc, CDP-Glc, ADP-Glc or TDP-Glc. Within the GT-A families, UDP-Glc is the predominant donor, however Family 81 enzymes utilize ADP-Glc as a donor. The nucleotides for the respective sugars should be mentioned in supplementary Figure 5. The modest predictive power for the unknown glycosyltransferases, with 45% low confidence are not unexpected given the lack of biochemical data for the training set which cannot distinguish between 7 sugars.

Taken together this is a novel contribution that will undergo continual refinement and updating as structures and functions GT-A enzymes are confirmed.

Reviewer #2:

Taujale et al., perform an extensive evolutionary analysis via sequence and structural (when available) comparisons of over 600,000 members of the glycosyltransferases GT-A fold superfamily. The authors generated a phylogenetic tree allowing them to cluster groups of GT-A fold families and subfamilies into clades and identify unique sequence features of subfamilies. They found that (1) glycosyltransferase function has been progressively diversified through both common core sequence mutations (donor specificity) and expansion of hypervariable loops (acceptor recognition), and (2) families exhibiting inverting and retaining mechanisms were found in the same clades suggesting that these catalytic mechanisms appeared independently many times during the evolution of the GT-A fold. Employing a machine learning framework trained with their data, they were able to predict known enzyme specificities with 92% accuracy and specificities of unknown glycosyltransferases with 55% confidence.

In sum, this is an extremely well written and well documented article that provides an evolutionary framework for the variation in GT-A fold glycosyltransferases, with specific and well-illustrated examples to support the authors' conclusions. Finally, their analysis of the new ML framework suggests that it will be an outstanding tool for the prediction of the specificity of unknown glycosyltransferase sequences. This manuscript and the resulting tool will be a great resource for glycobiologists and enzymologists.

Reviewer #3:

Summary:

In the paper, Taujale et al., use sequence mining in an attempt to understand the evolutionary basis behind the mechanism and diverse substrate specificity of glycosyltransferases (GT), specifically of the GT-A fold family. The authors analysed over 600,000 GT-A sequences from which they identified and defined a conserved minimal GT-A core and assembled a phylogenetic tree in order to gain insights into the evolution of the retaining and inverting mechanisms. Furthermore, machine learning was used to predict donor specificity with ~ 88% accuracy.

The author's analysis reveals new clustering of GT-A sub families, previously undescribed by CAZy and their machine learning model gave insights into features contributing to donor specificity. Overall, the method shows promise as a useful strategy to help guide the characterization of GT-As and provides a resource to derive new hypotheses on the functions of different GT-As. It could become acceptable if the authors can adequately address these points.

Major comments:

- It would be helpful to introduce a figure showing the inverting and retaining mechanisms into the introduction to clarify the discussion for the reader. Likewise, a figure showing the three GT folds as well as the relative abundance of these folds would place the discussion in context.

-Introduction: Should be 109 GT families, not 109 GT-A families (actually it is now 110). Additionally, it would benefit the readers if the number of GT-A families present in CAZy, be included.

-The authors do not reference or discuss an important paper (Yang et al., (2018)) that is closely related. In this, an informatics approach was applied to a large set of GT1s from *Arabidopsis thaliana* and used to predict substrate specificities by training a set with data derived in the lab. They had considerable success in their predictions. Unfortunately, these are all GTB enzymes…so their results cannot be dropped into the current analysis…but the approach and their results should be discussed.

- Likewise, it would be interesting for the authors to analyze directed evolution experiments on the GT-A family and confirm that altered substrate specificity does indeed follow from mutations in the hypervariable loops. Such GT engineering experiments have been reviewed by Hancock et al., (2006) and more recently by McArthur et al., (2016).

-Discussion section: The authors should explain the 'Red Queen' effect for readers who might not be familiar with the analogy.

-Subsection “Machine learning analysis”: It would be interesting to know how the 713 characterized GT-A domains used for ML training as well as the test set are distributed in the Phylogenetic tree (Figure 2).

- The authors should reference, and perhaps provide a comparison, to the machine learning approach for GT1 activity predictions.

---

## [Author Response]

Summary:The basis for the diverse donor and acceptor substrate specificity of glycosyltransferases (GTs) was probed by global sequence analysis. Deep mining of 600,000 sequences of glycosyltransferase members of the GT-A fold family defined a minimal common core and hypervariable regions for these functionally diverse enzymes. Variations in these regions contribute to diversification of substrate specificity. A phylogenetic tree was generated from a representative set of GT-A sequences that showed retaining and inverting mechanisms evolved independently multiple times. The analysis revealed new clustering of GT-A sub families and clades, previously undescribed by CAZy. Machine learning was used to predict donor sugar specificity addressing a major problem for glycosyltransferases given the low numbers of biochemically and functionally characterized GTs.The major conclusions and the design of the study are of sufficient interest for publication in eLife. The study design goes beyond a simple evolutionary mapping, documenting supporting evidence for their conclusions with selected enzymes and using data to create a tool for predicting substrate use.The manuscript is well written requiring editorial correction as noted in the reviewers' minor comments. It would be nice to include figures showing retaining and inverting glycosyltransferase mechanisms and examples of the three glycosyltransferase folds with their abundances for general eLife readers. Explain the Red Queen effect for readers unfamiliar with the analogy. Prior analysis of Arabidoposis family 1 members comprised of the GT-B fold should also be discussed including comparison of the machine learning approaches. Analysis of experiments on the directed evolution of GT-A members would provide additional support for the suggestion that alterations in substrate specificity arise from hypervariable loop mutations. The distribution of the 713 members of the test set within the phylogenetic tree is also of interest.We have addressed these points in the revised manuscript. Our responses to each of these comments are provided below.Reviewer #1:This is a much needed analysis of the entire GT-A fold family of glycosyltransferases (GTs) focusing on their conserved structural features and evolutionary relationships within the framework of their CAZy defined families. Key findings are the identification of conserved secondary structures and hypervariable regions characteristic of the GT-A family including a common core with hydrophobic and active site DXD/xED metal binding residues, a G-loop and C-His. The core alignment was used to generate a phylogenetic tree for a subset of representative sequences.The phylogenetic tree defines sub-families within large CAZy GT families (GT2, GT8 and GT31) and groups GT families and sub-families into clades. This is a significant achievement given that the lack of sub-groups within the large GT families and identification of clades is a limitation of the CAZy database.An intriguing feature of the phylogenetic tree was the location of mechanistically distinct inverting and retaining enzymes throughout the identified clades. This suggests that retaining and inverting enzymes appeared independently multiple times through GT-A evolution.The prediction of donor sugar specificity addresses a major problem for glycosyltransferases given the low numbers of biochemically and functionally characterized GTs. This was based on a machine learning of donor types, however, this prediction is for the sugar transferred from the donor and not the nucleotide. For example, glucose is transferred by different GTs from UDP-Glc, CDP-Glc, ADP-Glc or TDP-Glc. Within the GT-A families, UDP-Glc is the predominant donor, however Family 81 enzymes utilize ADP-Glc as a donor. The nucleotides for the respective sugars should be mentioned in supplementary Figure 5.

The nucleotides for the sugars have been added to the labels in Figure 6—figure supplement 1.

The modest predictive power for the unknown glycosyltransferases, with 45% low confidence are not unexpected given the lack of biochemical data for the training set which cannot distinguish between 7 sugars.Taken together this is a novel contribution that will undergo continual refinement and updating as structures and functions GT-A enzymes are confirmed.

We have implemented new machine learning methods to improve the prediction accuracy. We have also used additional analyses to find the most contributing features for each sugar donor in the machine learning model.

Reviewer #3:Summary:In the paper, Taujale et al., use sequence mining in an attempt to understand the evolutionary basis behind the mechanism and diverse substrate specificity of glycosyltransferases (GT), specifically of the GT-A fold family. The authors analysed over 600,000 GT-A sequences from which they identified and defined a conserved minimal GT-A core and assembled a phylogenetic tree in order to gain insights into the evolution of the retaining and inverting mechanisms. Furthermore, machine learning was used to predict donor specificity with ~ 88% accuracy.The author's analysis reveals new clustering of GT-A sub families, previously undescribed by CAZy and their machine learning model gave insights into features contributing to donor specificity. Overall, the method shows promise as a useful strategy to help guide the characterization of GT-As and provides a resource to derive new hypotheses on the functions of different GT-As. It could become acceptable if the authors can adequately address these points.Major comments:- It would be helpful to introduce a figure showing the inverting and retaining mechanisms into the introduction to clarify the discussion for the reader.

We have added a new Figure 1 where the lower panel shows the well accepted mechanism of transfer for inverting and retaining GTs. Only the S_N_i transfer mechanism has been shown for the retaining GTs since this has been the most widely accepted in the field. However, as the reviewer has also suggested, the retaining mechanism for the GTs is still not clear and other mechanisms for retaining GTs have also been proposed which have been reviewed in [Moremen et al., (2019)], and are not included in Figure 1 for brevity.

- Likewise, a figure showing the three GT folds as well as the relative abundance of these folds would place the discussion in context.

We have added Figure 1 top panel which shows representative structures for the 3 main GT folds – GT-A, GT-B and GT-C in an orientation that clearly highlights the differences between the 3 folds. Figure 1—source data 1 includes a table that lists all the CAZy numbered families from 1 to 110, their assigned 3D structural fold and the number of sequences in Archaea, Bacteria and Eukaryota for that family. This table could serve as a helpful resource for researchers looking to summarize the distribution of sequences across the 3 folds and CAZy families.

-Introduction: Should be 109 GT families, not 109 GT-A families (actually it is now 110). Additionally, it would benefit the readers if the number of GT-A families present in CAZy, be included.

This has been corrected to 110 GT families.

-The authors do not reference or discuss an important paper (Yang et al., (2018)) that is closely related. In this, an informatics approach was applied to a large set of GT1s from *Arabidopsis thaliana* and used to predict substrate specificities by training a set with data derived in the lab. They had considerable success in their predictions. Unfortunately, these are all GTB enzymes…so their results cannot be dropped into the current analysis…but the approach and their results should be discussed.

We thank the reviewer for pointing us to this very important paper. As stated, Yang et al., applied their approach on a GT-B fold family which makes direct comparisons difficult. Nevertheless, based on approaches presented in the Yang et.al., paper, we have implemented a gradient-boosted regression tree classifier method for donor prediction. Based on this new method, we are able to achieve nearly 90% prediction accuracy. These additions are now incorporated in the revised methods, Results section and Discussion section of the manuscript. Per the reviewer’s suggestion, we now discuss our machine learning predictions in light of previously published directed evolution experiments.

- Likewise, it would be interesting for the authors to analyze directed evolution experiments on the GT-A family and confirm that altered substrate specificity does indeed follow from mutations in the hypervariable loops. Such GT engineering experiments have been reviewed by Hancock et al., (2006) and more recently by McArthur et al., (2016).

The directed evolution experiments discussed in Hancock et al., and McArthur et al., provide context for donor sugar specificities in GT-A fold enzymes. The mutations discussed in these reviews support the important features identified for donor binding from our machine learning analysis and have been used to further substantiate these findings. These reviews and relevant papers are discussed in the subsection “Machine learning” and in the Discussion section.

For the hypervariable loops and their roles in substrate binding, we now cite multiple papers that report specificity determining residues and conformational changes induced by substrate binding. Again, we thank the reviewer for pointing us to these important reviews that have been very helpful in analyzing and interpreting the results from our machine learning analysis for donor substrate specificity.

-Discussion section: The authors should explain the 'Red Queen' effect for readers who might not be familiar with the analogy.

We have provided a layman’s explanation of the ‘Red Queen’ effect in the revised Discussion section.

-Subsection “Machine learning analysis”: It would be interesting to know how the 713 characterized GT-A domains used for ML training as well as the test set are distributed in the Phylogenetic tree (Figure 2).

We have added a new Figure 6—figure supplement 1 which highlights the number of sequences from each of the major GT-A families that were present in both the training dataset and the prediction dataset. This figure provides a snapshot of how the training and prediction set sequences are distributed across the phylogenetic tree. We also provide a new Figure 6—source data 2 table which provides the specific number of sequences used in the training and prediction set.

- The authors should reference, and perhaps provide a comparison, to the machine learning approach for GT1 activity predictions.

We reference the Yang et.al., paper in the revised version (subsection “Machine learning to predict the donor specificity of GT-A sequences”, Discussion section) and acknowledged the use of similar decision tree based learning approaches for predicting GT functions. However, a direct comparison of our results with the Yang et al., study is not possible because the predictions are on two different folds and the scope of the training sequences are very different in the two studies.